# Apolipoprotein D in Oxidative Stress and Inflammation

**DOI:** 10.3390/antiox12051027

**Published:** 2023-04-28

**Authors:** Guillaume Fyfe-Desmarais, Fréderik Desmarais, Éric Rassart, Catherine Mounier

**Affiliations:** 1Laboratory of Metabolism of Lipids, Centre d’Excellence en Recherche sur les Maladies Orphelines-Fondation Courtois (CERMO-FC), Department of Biological Sciences, University of Quebec in Montreal (UQAM), 141 Av. du Président-Kennedy, Montreal, QC H2X 1Y4, Canada; desmarais-fyfe.guillaume@courrier.uqam.ca (G.F.-D.); rassart.eric@uqam.ca (É.R.); 2Department of Medecine, Faculty of Medecine, Institut Universitaire de Cardiologie et de Pneumologie de Québec, 1050 Av. de la Médecine, Québec City, QC G1V 0A6, Canada; frederik.desmarais.1@ulaval.ca

**Keywords:** arachidonic acid, metabolism, prostaglandins, obesity, alzheimer, high density lipoprotein, lipid peroxidation, polyunsaturated fatty acid, 4-hydroxynonenal, malondialdehyde

## Abstract

Apolipoprotein D (ApoD) is lipocalin able to bind hydrophobic ligands. The APOD gene is upregulated in a number of pathologies, including Alzheimer’s disease, Parkinson’s disease, cancer, and hypothyroidism. Upregulation of ApoD is linked to decreased oxidative stress and inflammation in several models, including humans, mice, Drosophila melanogaster and plants. Studies suggest that the mechanism through which ApoD modulates oxidative stress and regulate inflammation is via its capacity to bind arachidonic acid (ARA). This polyunsaturated omega-6 fatty acid can be metabolised to generate large variety of pro-inflammatory mediators. ApoD serves as a sequester, blocking and/or altering arachidonic metabolism. In recent studies of diet-induced obesity, ApoD has been shown to modulate lipid mediators derived from ARA, but also from eicosapentaenoic acid and docosahexaenoic acid in an anti-inflammatory way. High levels of ApoD have also been linked to better metabolic health and inflammatory state in the round ligament of morbidly obese women. Since ApoD expression is upregulated in numerous diseases, it might serve as a therapeutic agent against pathologies aggravated by OS and inflammation such as many obesity comorbidities. This review will present the most recent findings underlying the central role of ApoD in the modulation of both OS and inflammation.

## 1. Introduction

Apolipoprotein D (ApoD) is a small secreted glycoprotein (25–30 kD) belonging to the lipocalin family. It is implicated in oxidative stress (OS), inflammation, and the transport of small hydrophobic molecules. APOD has been found to be the gene that is most upregulated with age [1], and its expression increases in various pathologies, particularly in several neurodegenerative diseases such as Parkinson and Alzheimer [1,2,3,4,5] but also during OS, starvation/growth arrest [6,7,8,9,10,11,12,13,14], cancer [9,15,16,17,18,19,20], hypothyroidism [21], and acute inflammation in response to cerebral viral infection [22] and LPS [7]. This upregulation is most likely in response to the pathologies observed as ApoD seems to have a beneficial effect on the outcome of most of them.

This protein can bind a large variety of hydrophobic molecules but its preferential ligand is the C20:4 ω-6 polyunsaturated fatty acid (PUFA) arachidonic acid (ARA) [23]. Its affinity for ligands seems to differ depending on the organism in which the protein is expressed. For example, Neural Lazarillo (Nlaz), one of the two *Drosophila melanogaster* homologue of ApoD, can bind retinoic acid with a Kd of 1.4 +/− 1.6 μM, but when Nlaz is produced in bacteria, it binds retinoic acid with a Kd is 2.6 +/− 1.8 μM. The human ApoD can also bind retinoic acid, this one with a Kd of 4.0 +/− 2.6 μM [24]. Finally, ApoD has been associated with cholesterol with a weak or inapparent binding (Marcoux-Legault, Brissette and Rassart, unpublished results) [23,24,25]. Furthermore, the *Bombyx mori* homologue of ApoD (BmApoD1) does not bind cholesterol at all [26], but Nlaz and Glaz do [24]. More studies are required to confirm that ApoD can bind and/or transport cholesterol, as circulating ApoD is mostly associated with high-density lipoproteins [27,28]. Other ligands include but are not limited to steroids (progesterone and pregnenolone) [29,30], palmitic acid, sphingolipids, sphingomyelin [31], bilirubin and E-3-methyl-2-hexenoic acid (again, depending on the organism in which it is produced [32]) [33,34]. Importantly, ApoD can interact with reactive lipid hydroperoxides and act as an antioxidant, reducing the 5s-, 12s-, and 15s-H(p)ETEs derived from ARA [35] via its methionine 93 residue [36] (see Section 2.5: In silico studies).

Since ApoD’s expression is upregulated in numerous pathologies in relation with OS and inflammation, this review will present the most recent findings underlying a central role for ApoD in the modulation of both oxidative stress and inflammation.

### 1.1. The Protein

ApoD is a circulating glycoprotein, and its concentration in the human plasma varies between 28 ug/L and 5–20 mg/dL, depending on multiple factors such as age and pathology [37,38,39,40]. Isolated in 1973 from high density lipoprotein (HDL), the protein was named apolipoprotein D [41], but its sequence similarity, its expression pattern as well as its structure show that ApoD belongs to the lipocalin family. ApoD has actually been found to be one of the oldest lipocalins, and is present among almost all phyla [42]. Homologues of the human protein have been at least partially characterized in rats [43], rhesus monkeys [44], rabbits [45], Gram-negative bacteria [46], mice [47], *Arabidopsis thaliana* [48], *Drosophila melanogaster* [49], and in *Bombyx mori* [26]. Most of the homologues across different taxa appear to display similar roles.

In humans, the highest ApoD expression has been detected in the mammary gland, brain, spinal cord, and adipose tissue (Figure 1), while in rodents, it is mostly expressed in the central nervous system (CNS) [47], principally by glial cells (astrocytes and oligodendrocytes) [50]. Its expression in rodents has also been detected in the mammary gland, salivary gland, testicles and skeletal muscle, but at a lower level than in the CNS [51]. The ApoD transcriptional expression is tightly regulated in a context-dependent manner. In fact, our group demonstrated that the ApoD promoter contains numerous response elements such as NF-kB, AP-1 and APRE-3 response element, a progesterone and a glucocorticoid response element as well as binding sites for the transcription factors SP-1 and sterol-dependent repressor. A fat-specific element (FSE), an alternating Purine-Pyrimidine stretch (APP) as well as GC boxes were also identified [7,52,53]. The pathway involved in the upregulation of ApoD’s expression seems to be mainly mediated by the JNK signaling pathway, as inhibition of the JNK pathway during stress totally abolishes ApoD expression [54,55]. Interestingly, it has also been shown that Apolipoprotein E (ApoE), an apolipoprotein involved in lipid and cholesterol transport and Alzheimer’s disease, specifically the isoforms E3 and E4, can bind and repress ApoD expression [56].

Since ApoD is secreted, its presence as a protein in tissues is often different from the place where it is transcriptionally expressed. In humans, high ApoD protein levels were detected in adrenal glands, bronchus, kidneys, liver, placenta, and testis, where its mRNA level is almost not detected (Figure 1). Furthermore, our group showed that radiolabelled ApoD injected in the cerebral ventricles of mice can exit the brain and accumulate in many organs, primarily in the kidneys and liver, where it is poorly expressed. This tissue accumulation correlates with the protein level and degree of glycosylation of basigin (BSG), a receptor identified to be able to internalize ApoD [57,58]. In addition, compared to peripheral tissues, ApoD in the brain was found to be hypoglycosylated [59]. Since glycosylation impacts ApoD binding capacities [32], its conformation [36], and its stability in the lysosome [60,61], different glycosylation degrees might also play a role in ApoD distribution.

The spatial expression of ApoD is therefore variable and dependent on the cellular context. It appears that the common factor governing the expression of ApoD is OS and inflammation. In fact, most if not all of the pathologies in which ApoD’s expression is increased present an increase in OS and inflammation.

### 1.2. ApoD Internalisation and Intracellular Localization

Despite the fact that the exact internalization process of ApoD has yet to be clearly demonstrated, studies using different cells (HEK293T, 1321N1, U87, NIH/3T3 and primary murine astrocytes [7,55,60,62]) and mice model [58] demonstrate ApoD’s potential to be internalize and/or trafficked.

ApoD, at least in HEK293T and BSG-overexpressing SH-5Y5Y, is internalized in a basigin-dependent manner [57]. New results suggest its internalization does not always require basigin in U87 and 1321N1 cells [62]. After internalization, our group showed that ApoD accumulates in the nucleus of NIH/3T3 cells under stress conditions such as 6 days following LPS treatment or when cells are cultured in low serum, mimicking the growth arrest [7]. In HT22 mouse hippocampal neuronal cells, ApoD was found to translocate from cytoplasm to cellular extensions, and to the nucleus when treated with hydrogen peroxide (H2O2). The localization of ApoD in the nucleus was only present in cells showing nuclei morphological changes due to damage caused by OS. Therefore, the presence of ApoD in the nucleus might be due to the fact that the nuclear envelop is damaged, which would allow proteins to go inside the nucleus without the need for controlled nuclear translocation [63]. However, treatment with the ROS-generating agent paraquat (PQ) generated different results. In primary astrocytes purified from wild-type (WT) and ApoD-KO mice treated with PQ, exogenous HApoD is found in the perinuclear space, the intracellular membrane, and the vesicular compartments, but not in the nucleus [55]. In addition, it was recently demonstrated that following internalization, ApoD is trafficked to early endosomes and lysosomes. Upon OS, ApoD then colocalize with a specific subtype of lysosomes sensitive to OS and autophagosomes, at least in 1321N1 and fibroblastic cells [60,64]. Bacterially produced ApoD (unglycosylated ApoD) is also internalized and subsequently targeted to the lysosomes where it is rapidly degraded [60]. However, incubation of HEK293T cells with purified human ApoD (HApoD), showed that the protein is internalized and stable for over 50 h in the lysosomes, implying that ApoD’s glycans are necessary for its stability inside the lysosome [60].

Therefore, depending on the cells (BSG-KO U87, U87, NIH/3T3, HT22, HEK293T, and primary murine astrocytes) and/or the context (OS, inflammation, or growth arrest), the cellular distribution of ApoD might differ, making it hard to attribute a definitive localization and role.

### 1.3. The Main ApoD Ligand: Arachidonic Acid

Of great interest, ApoD’s preferentially binds the ω-6 polyunsaturated fatty acid (PUFA) arachidonic acid (ARA) [23,24], the main precursor of many bioactive lipids that are used as signaling molecules, with several being implicated in the inflammatory responses [65].

Once ARA is liberated from the plasma membrane via the activity of the phospholipase A-2 (PLA2) enzyme, it can be metabolized by two mechanisms: enzymatically and non-enzymatically (Figure 2). The enzymatic oxygenation of ARA can be catalysed by cyclooxygenases (COX), lipoxygenases (LOX) or cytochrome P450 (CYP450). This will generate the prostaglandins and thromboxanes (COX pathway), the hydroxyeicosatetranoic acids (HETEs) leukotrienes, and lipoxins (LOX pathway), and epoxyeicosatrienoic acid (EET) (CYP450 pathway) (Figure 2). The non-enzymatic transformation of ARA is either a lipid peroxidation (LPO) reaction which can be carried out by the interaction of ARA with lipid peroxides, generating the isoprostanes (also called PGF2-like compounds), or an oxidation reaction with various reactive species such as the *reactive oxygen species* (ROS), *reactive nitrogen species*, or *reactive carbonyl species* generating the isoprostanes and the isoleukotrienes (for an in-depth review of arachidonic acid metabolism, see [66,67]). Importantly, the LPO and oxidation of ARA can be deleterious, as it can generate aldehydes, such as the cytotoxic reactive carbonyl species 4-hydroxynonenal (4HNE) and malondialdehyde (MDA) [68,69]. Although no direct molecular interaction between these specific reactive carbonyl species and ApoD has been shown, many studies on ApoD use the aldehydes 4HNE and MDA as OS markers, both of which are consistently decreased when ApoD is increased (see Section 2).

It has been suggested that ApoD can sequester ARA either in the cell membrane and/or in the cytosol [70,71]. These suggestions have been supported by the fact that in first-episode neuroleptic-naïve schizophrenia patients, ApoD’s plasma level is correlated with red blood cell membrane ARA, but is not actually increased in the plasma compared to healthy controls [38]. It was also demonstrated that the absence of ApoD increases the level of free ARA in the intact nerve of ApoD-KO mice [72], supporting the idea that ApoD has a role to play in the control of release or stabilization of membrane-bound ARA. Therefore, knowing that ARA is prone to oxidation and LPO reactions, the role of ApoD in reducing the level of toxic ARA metabolites resulting from LPO (such as 4HNE and MDA) could be explain at least in part by its capacity to bind and sequester ARA.

It has also been shown that ApoD has an impact on PLA2 and COX-2. Indeed, overexpression of human ApoD in mice during acute cerebral inflammation decreases PLA2 activity, while overexpression of human ApoD in mice decreases COX-2 expression during OS in the hippocampus [22,73]. ApoD-KO mice also have a higher basal level of PLA2 activity than their WT counterpart in the intact sciatic nerve [72]. Finally, we have recently demonstrated that ApoD can alter the hepatic metabolism of ARA, increasing the hepatic level of ARA and its anti-inflammatory metabolites, PGD_2_ and 15d-PGJ_2_ [74]. Since the first step in ARA metabolism is the PLA2-mediated release from the cell membrane followed by the COX-2 generation of PGs, modulation of the activity of these enzymes should have an impact on inflammation [67]. Furthermore, the fate of ARA metabolism is subsequently dictated by the expression and activity of enzymes downstreams of PLA-2 or parallel to COX-2 (such as prostaglandin and thromboxanes synthases, lipoxygenases, CYP450). By modulating one or more of those enzymes, ApoD can hence modulate the downstream metabolism of ARA, and limit OS and/or inflammation. For example, ApoD could potentially also modulate the activity and/or expression of other enzymes, such as the LOX enzymes, to increase the synthesis of some anti-inflammatory specialized pro-resolving mediators, such as the *lipoxins*, further increasing the anti-inflammatory potential of ARA-derived metabolites (see Section 3, Modulation of Obesity-Related Inflammation). These mechanisms remain theoretical and should be examined.

## 2. Role of Apolipoprotein D in Oxidative Stress

The oxidative state of cells is controlled by a balance between the antioxidant entities (either enzymes, such as superoxide dismutases (SODs), or low-molecular-weight molecules, such as glutathione) and oxidative molecules, such as ROS, reactive nitrogen species, and reactive carbonyl species. These different entities are the key players in the reduction–oxidation reactions. Damage to the cell membranes, proteins, and nucleic acids (DNA and RNA) can occur when the balanced is tipped in favor of the oxidant molecules. This balance can be restored towards a physiologically balanced oxidative state via antioxidant mechanisms [75]; however, when uncontrolled, the balance is tipped in favor of the oxidants creating a state of a physiologically toxic oxidative stress [76]. This is thought to be one of the main factors involved in aging [77], and cellular OS can generate inflammation through various mechanisms such as the activation of the inflammasome [78]. OS can also contribute to the severity of neurodegenerative diseases such as Alzheimer’s disease (AD) [78,79], and can also contribute to obesity-related comorbidities such as non-alcoholic fatty liver disease (NAFLD) [80]. However, oxidation of biomolecules is not always deleterious and is sometimes used to produce bioactive molecules of various functions. For example, thiol-containing proteins sometimes require redox reactions to ensure correct protein folding in the ER via protein disulfide isomerase [81], while oxidation of the omega-3 fatty acids docosahexaenoic acid (DHA), docosapentaenoic (DPA) or eicosapentaenoic acid (EPA) by the LOX enzymes give rise to the anti-inflammatory lipid mediators called *resolvins, protectins and maresins* [65,82].

Since ApoD appears to contribute to the oxidative state of the cell in an antioxidant manner to decrease oxidized lipids specifically, but not proteins [8], most of our review will focus on the specific role of ApoD on oxidation of lipid species. For an exhaustive review concerning the role of ApoD outside of oxidative stress and inflammation, please see [83,84].

### 2.1. Human Studies

Few studies have identified a link between ApoD and oxidative stress in humans. Alzheimer’s disease (AD) is a progressive neurodegenerative disease caused by an abnormal deposition of amyloid B peptides and intracellular accumulation of neurofibrillary tangles of Tau proteins. This can be enhanced by oxidative stress, worsening the progression of the disease [85]. ApoD’s involvement in Alzheimer disease has been known for almost three decades, when it was first observed that its level in the cerebrospinal fluid and hippocampus of Alzheimer’s patient was more than 3× higher than normal [86,87,88]. The hippocampus’ role in learning and memory makes it a pivotal region of the cortex to study in AD. Furthermore, since oxidative stress is a major source of progression of AD [89,90] and ApoD acts in an antioxidant manner and is upregulated in this neurodegenerative disease, studies on the behavior of ApoD in this specific region of the brain are of interest. Analysis of the hippocampus of patients who had late-stages of AD reveal the presence of ApoD dimers, which are positively correlated with the levels of conjugated dienes [88], a lipid peroxidation marker. ApoD in its monomeric form uses its methionine 93 (Met93) to reduce lipid peroxides [35]. In turn, the oxidation of Met93 gives rise to the ApoD-Met93-SO form, which has a proclivity to form dimers via disulfide bonding of its residue Cys116 [91,92]. Monomeric ApoD-Met93-SO could hypothetically be returned to ApoD-Met93 by the action of methionine sulfoxide reductase enzymes. It is therefore not surprising that more ApoD-dimers is observed in the brains of patients in the late-stages of AD in which a reduction in the methionine sulfoxide reductase enzymatic activity is also observed [93], although no studies have considered possible molecular interactions between the two. Since a major component of the brain is lipids, particularly PUFAs, which makes it susceptible to lipid peroxidation and its toxic metabolites, reducing the amount of lipids prone to peroxidation by ApoD could reduce oxidative stress and protect the brain against the progression of AD [94].

Our group also showed that ApoD’s expression was decreased during normal uncomplicated pregnancy, and that ApoD’s levels were further decreased in women with a high gestational weight gain [95]. The decrease in ApoD during pregnancy might be due to the inhibitory effects of estradiol on ApoD’s expression [96]. Another group found that ApoD’s expression is increased in the placenta of women with gestational diabetes mellitus compared to healthy women, and it was suggested that the increase was possibly a mechanism in response to OS, although there was no correlation between the level of 4HNE and ApoD’s increase [97].

Finally, although a direct link was not observed, ApoD’s expression or protein level in humans is modulated in several pathologies such as Parkinson [3], Alzheimer [4], type 2 and gestational diabetes [97,98], Tangier’s disease [99], and various cancers [9,15,16,17,18,19,20]. All these diseases have an increase in OS, and more studies carried out in humans are required to determine the exact role of ApoD and oxidative stress in humans.

### 2.2. Animal Studies

Although new studies are now starting to focus on the molecular aspects of ApoD, most of the studies linking ApoD’s role in OS were carried out in vivo with animal models (mostly mice and *Drosophila melanogaster*), making connections between ApoD and the level of certain OS markers.

#### 2.2.1. Rodents

In ApoD-null-APP-PS1 mice, models of amyloidogenic AD without ApoD expression, loss of ApoD increases the amyloid plaque load by almost two-fold, while overexpression of human ApoD (HApoD-APP-PS1 mice) decreases by 60% the amyloid-β 1-40 peptides and decreases by 34% the insoluble amyloid-β 1-42 [100]. The Aβ (1-42) peptides are usually the ones associated with the highest increase in OS [101], and Aβ (1-40) are usually the most prevalent, but both cause amyloidogenesis and contribute to the progression of the disease [85].

In the same study, the authors showed that the expression of the BACE1 enzyme, responsible for cleaving the amyloid-beta precursor protein (APP) and enhancing amyloidogenesis [102], was increased in the ApoD-null-APP-PS1 mice and that, while not significant, the expression of the BACE1 protein was slightly decreased in HApoD-APP-PS1 mice. This suggests an inverse relation between the expressions of ApoD and BACE1 proteins. Since oxidative stress is associated with an increase in BACE1 enzyme expression [103], ApoD’s decrease in OS could indeed have an impact on BACE1 expression. Interestingly, LPO markers in these mice (as evaluated by measuring the levels of 4HNE and F2-isoprostane) remained unchanged regardless of the genotype, although a non-significant trend of increase/decrease for both OS markers depending on the genotype was observed (ApoD-null-APP-PS1 mice/HApoD-APP-PS1 mice, respectively) [100].

A study carried out on transgenic mice constitutively expressing the human ApoD in the brain under the human THY promoter (HApoD-Tsg) generated by our group and ApoD-KO mice were subjected to intraperitoneal injections of PQ. HApoD-Tsg mice showed a better survival rates than wild-type and the survival rate of ApoD-KO mice was much lower. In control conditions, lipid peroxidation in the brain, but not in lungs, was higher in ApoD-KO mice, while protein oxidation remained unchanged. Peroxidation of lipids in the brain was also higher in ApoD-KO compared to WT mice, only two hours after PQ injection as well as after two weeks of chronic low dose administration and protein oxidation remained unchanged. Overexpression of HApoD predictably decreased lipid peroxides levels when compared with wild type upon a two weeks of chronic low dose PQ administration. In fact, overexpressing the human ApoD actually prevented any increase in LPO in the brain over these two weeks [8] demonstrating the importance of ApoD in preventing this deleterious process.

ApoD also has an effect on the transcriptional response in the brain. ApoD-KO mice’s cerebellum have differential expression of genes related to inflammation and OS, as *MAP3k7* and *Nf1a* are upregulated, while *Ccl21* is downregulated compared to WT mice, suggesting an injury phenotype in control condition in ApoD-KO mice, as all three genes are related to oxidative stress and inflammation [104]. This might be connected to their elevated level of basal lipid peroxides [8], which can trigger a response to OS [105]. PQ-challenged HApoD-Tsg mice’s cerebellum have a modified transcriptional response to OS when compared to WT mice by having an almost completely abolished early OS transcriptional response [104]. While ApoD-KO might be more susceptible to OS, the fact that ApoD reduces OS markers could mean that overexpressing ApoD mice are protected from OS up to a point, thus having no need for an early transcriptional response to PQ.

Finally, ApoD has been shown to have antioxidant proprieties in mouse models of atherosclerotic coronary artery disease. SR-BI/ApoE double KO (dKO) mice develop coronary artery diseases, in which ApoD’s expression in the heart is increased by 80-fold [106]. Adenoviral-mediated hepatic overexpression of ApoD followed by ischemia and reperfusion of the heart showed a decrease in infarct size compared to empty-vector WT mouse. Predictably, loss of ApoD (ApoD-KO) mice had an increase in infarct size. The authors also found that ApoD can protect primary rat cardiomyocyte from hypoxia/reoxygenation injury. Since all these situations deeply involve OS, the authors investigated whether ApoD had the antioxidant potential to inhibit the oxidation of 2,2′-azino-bis(3-ethylbenzthiazoline-6-sulfonic acid) diammonium salt (ABTS) by hydrogen peroxide and myoglobin. Remarkably, ApoD was an even more potent antioxidant than Trolox (1.2 µmol/L apoD was equivalent to 314 µmol/L Trolox), a vitamin E derivative, further expanding our knowledge of its capacity to reduce a wide array of oxidant molecules, not only lipids [106].

#### 2.2.2. Insect Models

Other ApoD homologues have also been shown to ameliorate the oxidative state of the organism in which it is expressed. Although their exact amino acid sequence does not always match, and not all have the conserved Met93 observed in all mammals, many studies show that ApoD’s insect homologues expression in response to OS is upregulated.

Several studies have been carried out in the *Drosophila melanogaster* model on human ApoD and its ApoD insect homologues, Neural Lazarillo (NLaz) and Glial lazarillo (GLaz) to look at the effect of loss of function or overexpression of these proteins on OS. GLaz-KO Drosophila challenged with PQ, hydrogen peroxide or starvation have a lower survival rate than their WT counterpart [107] while inversely, *D. melanogaster* overexpressing GLaz under hyperoxia or starvation have a higher survival rate [108]. Similarly, when drosophila-overexpressing NLaz are starved, an increase in the survival rate is observed [109]. All these challenges create some forms of OS. Predictably, overexpression of GLaz or HApoD in *D. melanogaster* reduces the level of MDA and 4HNE that accumulates with age [110,111]. The inverse is also true in GLaz-KO flies, which accumulate more MDA over time than their wild-type counterparts [107].

Some differences have been noted between the effects of Nlaz and Glaz. Nlaz has been proposed to be implicated in metabolism, while Glaz has been proposed to be implicated in the aging brain; however, both reduce lipid peroxides and respond to OS. Furthermore, abrogation of Glaz decreases longevity in males but not in female, while abrogation of Nlaz decreases longevity in both male and female [112]. All ApoD homologues display similar roles, as overexpression of the grasshopper *Schistocerca americana* Glaz, Nlaz and ApoD homologue, *Lazarillo* (Laz), also extends lifespan in *D. melanogaster*. The overexpression of HApoD also increases the lifespan of these flies [110].

In a *D. melanogaster* model of Friedreich’s ataxia, the most common form of autosomal recessive ataxia is caused by the loss of mitochondrial frataxin. In this situation, an increase in OS is observed associated with elevated LPO and subsequently neurodegeneration, and impaired locomotor activity. Co-expression of GLaz in glial frataxin deficient *D. melanogaster,* partially restores the effect of frataxin loss on survival, locomotor activity, and LPO [111]. Loss of glial frataxin expression also induces increase in free fatty acid (FFA) and triglycerides content in flies which is reduced by GLaz co-expression. Specifically, GLaz co-expression reduces C16:1 (palmitoleate), C18:1 (oleate) and C14:1 (myrestoleate) accumulation, but does not affect the other fatty acids tested in this study [111], pointing towards a potential role of Glaz in lipid metabolism. Remarkably, co-expression of Glaz was also able to restore the activity of aconitase (an enzyme susceptible to loss of function upon OS) to the level observed in WT flies. The restoration of aconitase activity suggests once again that GLaz is strongly linked to the control of the balance between pro- and antioxidants.

Overall, similarly to ApoD, Laz, GLaz and NLaz all seem to act as antioxidant proteins, reducing LPO.

In a study carried out with the lepidopteran model *Bombyx mori*, the ApoD homologue’s (BmApoD1) expression was upregulated upon H_2_O_2_ challenge, starvation, and bacterial challenge in *B. mori* larvae [11,26]. Exogenous BmApoD1 addition to BmN cells (*B. mori* ovary cells) also increased survival upon challenge with 1mM H_2_O_2_ treatment and also inhibited actinomycin D-induced apoptosis [26]. The mechanism preventing cell death upon Actinomycin-D treatment might be linked to the reduction in lipid oxidation as treatment of HEK293T and LO-2 cells with actinomycin-D increases LPO, which is decreased by ApoD overexpression in many cell types [113]

### 2.3. Plant Studies

Plants also have lipocalins that share a significant sequence similarity with ApoD. The homologues of ApoD in *Arabidopsis thaliana* are AtTIL and LNCP (previously named CHL for chloroplast lipocalin). It was shown that overexpression of AtTIL provides an increase in the survival of *A. thaliana* if the plants are subjected to freeze stresses, while AtTIL-KO *A. thaliana* showed lower survivability. Freeze stress without acclimatation in plants creates acute OS, suggesting that AtTIL is a key player in the acute antioxidant response in *A. thaliana*. However, if AtTIL-KO plants are subjected to a freeze stress after a period of cold acclimatation, the difference in survival between AtTIL-KO plants and wild-type plants disappears, suggesting that AtTIL is not necessary when plants go through the full gradual process of low temperature acclimatation [114]. AtTIL-KO plants also have increased necrotic lesions due to PQ treatment when compared to WT, while AtTIL-overexpressing plants have decreased necrotic lesions.

AtTIL also seems to play a role in plant growth in relation with OS. If AtTIL-KO plants are grown in continuous, moderate intensity light, smaller cotyledons are observed, suggesting they cannot tolerate the OS generated by the continuous light. AtTIL-KO plants also do not survive when first grown under dark conditions and returned to a normal light-dark cycle, unlike WT and AtTIL overexpressing plants. Furthermore, 3,3′-diaminobenzidine (DAB) staining (measuring H_2_O_2_ levels) also show higher levels in AtTIL-KO plants. These results suggest that plants lacking AtTIL accumulate ROS in a deleterious way, affecting normal growth and development [114].

Recently, human ApoD was expressed in *A. thaliana* lacking the plant ApoD homologue LNCP (LNCP-KO). The authors showed that the expression of HApoD in the chloroplast of LNCP-KO *A. thaliana* could partially rescue the plants responses to drought stress. These HApoD-LNCP-KO transgenic plants were also challenged with PQ during an 8 h light period and the photosynthetic functions were examined. HApoD-LNCP-KO transgenic plants have a better tolerance to PQ than WT plants, but still less than the plants overexpressing LNCP alone. MDA level, measured via the Thiobarbituric reaction assay (TBARS), increased upon PQ treatment, and HApoD-LNCP-KO transgenic plants had a lower level of LPO than WT plants, but still less than LNCP overexpressing plants. Even though HApoD and LNCP share significant sequence similarities, the authors suggest that inappropriate post-translational processing of the human ApoD in the plant could contribute to the differences observed between LNCP overexpressing plants and HApoD-LNCP-KO transgenic plants. This could be due to the different level of ApoD glycosylation in plants, possibly preventing human ApoD’s import into the lumen of thylakoids, where LNCP is located [115]. This could also compromise the ability of ROS or LPO products to interact with ApoD’s residue responsible for the reduction in reactive molecules (Met93).

### 2.4. Ex Vivo and In Vitro Studies

#### 2.4.1. ApoD in the Hippocampus

In rat hippocampal slice cell culture, addition of human ApoD upon kainate injury (massively increasing ROS and LPO products) confers a protective effect, preventing the decrease in microtubule associated protein-2 (MAP-2), and decreasing cell death and the activity level of lactate dehydrogenase (LDH) [71], both markers of neuronal cell injury that decrease and increase, respectively upon kainate treatment. Furthermore, ApoD was able to decrease both F2-isoprostane and 7-ketocholesterol formation upon kainate treatment. The authors suggest that ApoD’s binding of ARA and cholesterol might prevent their oxidation and thus decrease the level of the peroxidation marker observed. Kainate induces liberation of ARA from the cellular membrane by activation of phospholipase A2 (PLA2) [116], while also increasing OS and LPO [117,118]. Addition of ApoD, which most likely sequesters ARA to the cell membrane, would indeed reduce the level of F2-isoprostane, a product of ARA LPO. As already mentioned, human ApoD’s capacity to bind cholesterol is weak, and might therefore not be enough to directly shelter it from peroxidation. However, ApoD’s antioxidant mechanisms could be upstream of the events responsible for 7-ketocholesterol formation, for example, by preventing an LPO chain or propagation of various ROS which are responsible for 7-ketocholesterol formation [119]. This mechanism has yet to be demonstrated.

In another study, it was also found in HT22 mouse hippocampal cells that ApoD’s protein level was increased in a dose-dependent manner upon H_2_O_2_ treatment, and that lipid peroxide level was also correlated in a dose-dependent way to H_2_O_2_ [10]. The authors thus inferred that ApoD protein level was correlated with lipid peroxide levels.

#### 2.4.2. ApoD Acts as an Antioxidant in a Paracrine, but Also Autocrine Way

Since ApoD is a secreted protein, and is mainly expressed in glial cells (primarily astrocytes and oligodendrocytes) [120], it is worthwhile to consider how ApoD is transported, how it is internalized, and what effect there is in an OS context. In 1321N1 astroglioma and SH-SY5Y neuroblastoma cells, it was found that ApoD derived from extracellular vesicles can be transferred from astrocytes to neurons, acting in a paracrine way to protect neurons from OS. When SH-SY5Y neurons were treated with PQ and incubated with either 1321N1 astroglioma conditioned media or extracellular vesicles isolated from 1321N1 media, cell viability increased. Interestingly, in SH-SY5Y neurons treated with PQ, the rescue in viability from addition of astroglioma-derived extracellular vesicles was similar to when purified ApoD was added, suggesting that ApoD is the main component of the extracellular vesicles responsible for the increase in viability. Meanwhile, ApoD-KO murine primary astrocytes subjected to PQ treatment present an increase in viability when incubated with WT murine primary astrocyte conditioned media, indicating that ApoD is also capable of acting in an autocrine manner [121]. This highlights the importance of ApoD in the central nervous system in the protection of various cell type from PQ-induced OS and cell death. ApoD secreted by astrocytes and oligodendrocytes can therefore act not only in an autocrine way, but also in a paracrine way on adjacent cells.

#### 2.4.3. Mouse Bsg-KO Glioblastoma Cells

Previous studies have identified Basigin (BSG) as one of the receptors responsible for ApoD’s intracellular internalization and whole body distribution in mice [57,58]. BSG-KO glioblastoma U87 cells have recently been used to verify the role of BSG for ApoD’s internalisation [62], but have also been used in parallel to evaluate the effect of ApoD on the level of LPO upon paraquat treatment. BSG-KO U87 cells present a lower basal level of 4HNE and higher level of ApoD mRNA than U87 cells. Treatment of BSG-KO U87 cells with PQ increases the level of 4HNE which is abrogated via the addition of exogenous ApoD.

BSG (also known as CD147) is also an accessory glycoprotein that is required for proper folding and trafficking of the lactate transporters MCT1 and MCT4. Loss of function of these lactate transporters pushes the cells resort to oxidative phosphorylation more than anaerobic glycolysis, increasing OS originating from the mitochondria [122]. It is interesting to see that a lack of BSG—which pushes these cells to resort to an almost exclusive oxidative phosphorylation mechanism for ATP generation—leads to a lower level of 4HNE. The authors argued that since the BSG-KO cells are required to use almost exclusively oxidative phosphorylation, and oxidative phosphorylation is a major generator of ROS, the cells would have a higher basal antioxidant defense mechanism. It was also pointed out that lactate accumulates in these cells, and that lactate might have antioxidants proprieties, thus decreasing the damage carried out by ROS and therefore the need for an increase in antioxidant response.

#### 2.4.4. ApoD in the Lysosome as an Antioxidant

Over time, the focus on ApoD’s paradigm has shifted from strictly a lipid or hydrophobic molecule transporter associated with HDLs to an antioxidant protein capable of protecting the lysosomes from OS. Lysosome’s membranes are sensitive to oxidative stress causing membrane injury, leading to permeabilization and loss of protons and therefore inadequate pH for enzymic activity as well as protein leakage into the cytoplasm. This can lead to dysfunctional macromolecule turnover, be it lipids, proteins or others, as observed in lysosomal storage diseases, some of which ApoD is involved in (Niemann–Pick disease type A and C) [64,123].

ApoD’s colocalization with a specific subtype of lysosome was recently discovered. Indeed, in U87 glioblastoma cells, and primary mice astrocytes and Schwann cells, ApoD colocalize with Lamp2, a marker for late-endosomal-lysosome marker, and with LC3, an autophagosome marker [60,61,62,121]. Furthermore, ApoD seems to be necessary to mitigate lysosome alkalinization (and regain of adequate pH after alkalinization) and reduce membrane permeability upon paraquat (PQ) injury in a Niemann–Pick disease type A model [60,61,64]. PQ treatment causes lysosome alkalinization and intracellular lysosome defect in trafficking, both of which are rescued by presence of ApoD and seem to be linked to prevention of lysosome membrane permeability. Upon PQ addition, lysosomes of ApoD-KO primary murine astrocytes, differentiated neuroblastoma SH-SY5Y cells and HEK293T (all of whom lack ApoD expression) will alkalize via proton leakage without returning to baseline pH, whereas addition of exogenous human ApoD will mitigate their PQ-induced alkalinization and return their lysosomal pH to baseline [60].

ApoD also seems to enter a specific subset of lysosomes who are more sensitive to OS. The endocytosis of ApoD in SH-S5Y5 in particular is also dependent upon oxidative injury. The protein colocalizes rapidly with Lamp2 after PQ addition, but enters SH-S5Y5 cells slowly if the PQ treatment is not applied. This is in contrast with HEK293T cells, where ApoD internalisation and presence in the cell is rapid and stable without PQ addition, peaking early after addition of exogenous human ApoD and staying stable for at least 50 h [60]. This could suggest that HEK293T have a high OS basal rate, and ApoD is required to deal with OS injury in the lysosomes of HEK293T, but could also suggest a differential role of ApoD in non-neuronal cells. It could also suggest that ApoD accumulates stably over time in these cells, as cerebral ApoD strongly accumulate in the kidney [58]. Studies on the stability of ApoD in different types of cells, particularly primary cells, are of interest, knowing that ApoD can accumulate in a variety of cells outside the nervous system [58].

ApoD’s expression is also linked to an increase in lipofuscin accumulation in the cortex of ApoD-KO mice. Lipofuscin is a pigment granule that results from unsaturated fatty acid oxidation and accumulates in lysosomes with age. Knowing that lipofuscin accumulation has been associated with lysosome and autophagy disfunction, LPO and OS (reviewed in [124]), and that ApoD positively affects all those processes, it is not surprising to see that mice lacking ApoD have increased lipofuscin accumulation with age in their cortex [125].

### 2.5. In Silico Studies

Analysis of the ApoD protein sequence showed that there are three methionine localized at the positions 49, 93 and 157 (Figure 3). Met93 is a highly conserved residue across mammals and is the agreed-upon amino acid needed for its antioxidant mechanism. As observed in Bathia et al.’s work, [35] incubation of ApoD with various H(p)ETEs leads to the formation of their non-reactive HETE counterpart. However, the substitution of Met93 for an alanine residue decreased the reduction of 15s-H(p)ETE to 15s-HETE by ApoD, while the other methionine substitutions did not significantly alter the conversion. The authors also observed that during the reduction of H(p)ETEs to HETEs, the Met93 is oxidized forming the methionine sulfoxide (MetSO) [35].

Molecular dynamics analysis with NAMD later revealed that Met157 is buried and not accessible for interactions with the H(p)ETES ligands and Met49 is shielded by the N-glycan moiety on Asn45 and therefore not accessible either, but that the 5s-, 12s- and 15s-H(p)ETEs do interact with ApoD by wrapping around the Met93 residue. The same simulation demonstrated that the presence of MetSO in the ApoD protein does not appear to induce a major conformational change in the protein, but does allow the side chain bearing MetSO to become more flexible, possibly favoring the dimerization and the aggregation of the ApoD protein [36], a conformation often observed in the brain of Alzheimer’s patients [88].

While the Met93 residue of mammalian ApoD appears to be essential to mediate the antioxidant capacities of ApoD, other ApoD homologs lacking Met93 still have antioxidants capacities. In fact, methionine93 is not present in all homologs of ApoD. For example, recombinant ApoD from Amphioxus (*Branchiostoma belcheri*), BbApoD, produced in *Escherichia coli*, has also been shown to act as a scavenger protein that reduces the level of hydroxyl radicals and prevents supercoiled DNA-nicking in a dose-dependent manner [127]. However, the Amphioxus ApoD protein has a different expression pattern from that of vertebrates. Specifically, and more importantly, it lacks the equivalent of Met93 [128]. No molecular mechanism has been proposed to explain the antioxidant capacities of BbApoD.

The identity of other oxidants molecules with which ApoD can directly interact with and reduce to their non-reactive form is not known. It would therefore be of great interest to identify them to help understand the antioxidant properties of ApoD and its physiological role.

## 3. Role of ApoD in Inflammation

Inflammation, or the inflammatory cascade, is a complex process that allows and controls the remodeling and repair of tissues after injury or infection. A very large number of molecules, cells and systems are implicated in the inflammatory process, and to describe them would be beyond the scope of this review (for more details, see the review in [129]). However, a role for ApoD in mediating inflammation has been characterized in the nervous system and in various pathologies, such as disorders associated with obesity.

Evidence for ApoD as a direct anti-inflammatory mediator is less clear than for its antioxidant proprieties. One can speculate that by acting as an antioxidant, ApoD could diminish the OS-induced inflammation. However, as we will see, ApoD could also act as an anti-inflammatory agent by modulating other pathways, notably the ARA metabolic pathway (see Figure 1).

### 3.1. Modulation of Inflammation in the Nervous System

ApoD has been shown to be implicated in the inflammatory process in the nervous system, particularly in regulating the activation of the immune system leading to cell infiltration and secretion of cytokines. ApoD expression has been shown to be increased in the regenerated and remyelinating sciatic nerves [43,72], and in various injuries of the nervous system associated with stroke [130,131], entorhinal cortex lesion [132], traumatic brain injuries [133], coronavirus OC43 encephalitis [22] and experimental kainic acid induced lesions [134].

Nerve crush models have been used to study the behavior of the immune system in mouse lacking ApoD. In a nerve crush model, the nerve undergoes Wallerian degeneration–regeneration (WDR), a process that uses immune cells to clear myelin and heal the injury. When the nerve is crushed, the distal part undergoes myelin degradation. Then, macrophages and Schwann cells infiltrate the site of injury and clear debris via phagocytosis stimulated by the Galectin-3 protein. Finally, once the debris are cleared, the nerve undergoes myelin regeneration for a period of time helped by the action of the immune system which secretes growth factors. Interestingly, it was shown that after nerve crush model in WT mice, ApoD colocalizes with S100 positive cells, a marker Schwann cells activation. When the ApoD expression is abrogated (ApoD-KO mice), the WDR process is impaired. This is associated with increased macrophages infiltrations in the nerves of ApoD-KO mice at 7 days post crush injury, which persists over time up to at least 48 days, unlike in WT mice, where macrophage infiltration diminishes over time and is normally resorbed within 48 days [72]. A significant difference in Galectin-3 and GAP-43 accumulation between ApoD-KO mice and WT mice was also observed as early as 14 days post crush injury in the injured nerve, with ApoD-KO mice having a higher level of Galectin-3 and a lower level of GAP-43 than WT [135]. ApoD-KO macrophages also have a decreased capacity in myelin recognition, phagocytosis, and clearance [72,135]. Increases in Galectin-3 and impaired myelin clearance both indicate a lysosomal dysfunction. Normally, a decrease over time of Galectin-3 indicates that phagocytosis and resolution of myelin clearance is over. In this situation, ApoD colocalizes with S100 positive myelin debris, increasing phagocytosis of myelin by primary peritoneal macrophages [72]. Other models have also shown that ApoD is crucial in the process of myelin phagocytosis and optimization of autophagy [60,136]. Finally, in ApoD-KO mice, the sciatic nerves display an increase in basal transcription and protein level of pro-inflammatory markers, mainly IL-6 compared to WT mice. In the same study, it was showed that in the nerve crush model, the sciatic nerve of ApoD-KO mice has an increased TNF-α and MCP-1 gene transcription [72]. All these altered responses point to a role of ApoD in the WDR process intimately tied to the immune system. In addition, ApoD influences the cytokine response; however, the molecular mechanism responsible for this is unknown. One possibility concerns the management of debris clearance. An absence of ApoD may provide an environment where the debris can be recognized via DAMP receptors, which can in turn trigger the upregulation of inflammatory cytokine genes. Studies on the effect of ApoD on microglial cells have recently been conducted. It was found that primary ApoD-KO microglial cells have a decreased secretory profile and an increased phagocytotic phenotype. On the other hand, acute exposure of BV2 microglial cells to ApoD induces a switch from a resting, non-secretory state to a secretory and less phagocytic state [137].

Galectin-3 was not only modulated during the nerve crush model experiment. Our grouped demonstrated that Mac-2 (Galectin-3) was also decreased in HApoD-Tsg in a model of kainate-induced neurotoxicity compared to WT. Furthermore, HApoD-Tsg mice also had a lower COX-2 expression, supporting the notion that ApoD alters ARA metabolism [73].

Infection with the HCov-OC43 virus causes an increase in ApoD’s expression and a powerful and deadly encephalitis associated with an important inflammation in mice. Interestingly, the level of the pro-inflammatory cytokines TNF-α and IL-1β is lower in HApoD-Tsg mice infected with HCov-OC43 compared to the level observed in WT mice. HCov-OC43-infected HApoD-Tsg mice also showed a reduction in both brain PLA2 activity and macrophage infiltration compared to the WT mice. Despite this, an increase in astrocytes and microglial cells apoptosis was observed during the acute phase of the infection in HApoD-Tsg mice but a return to normal levels occurs once the infection is cleared [22]. This suggests that ApoD allows the modulation of the immune system upon viral infection in different ways. The mechanism through which ApoD works is still unclear; however, by limiting PLA2 activity, ApoD could inhibit the production of pro-inflammatory eicosanoids from ARA. The link between ApoD, PLA2, and pro-inflammatory eicosanoids should be investigated further.

Metabolism of ARA by PLA2 will also activate COX-2 leading to the synthesis of pro-inflammatory prostaglandins. As mentioned earlier, the expression of COX-2 is reduced in HApoD-Tsg mice having undergone kainic acid induced neurotoxicity [73]. Therefore, by reducing COX-2 activity, ApoD can reduce the inflammatory process promoted by the prostaglandins. This suggests that ApoD has the potential to regulate inflammation upstream of the prostaglandin cascade of events. In addition, also in HApoD-Tsg mice having undergone kainate induced neurotoxicity, a reduction in Galectin-3 and GFAP (a marker of astrocytes activation) is observed, reinforcing the idea that ApoD modulates astrocyte reactivity.

### 3.2. Modulation of Obesity-Related Inflammation

Low-grade chronic inflammation is one of the hallmarks of obesity. The metabolic inflammation present in obese individuals has long been known to contribute to the development of obesity’s numerous comorbidities such as type 2 diabetes and steatohepatitis. [138]. Several factors have been identified as contributors to the generation of this low-grade chronic inflammation: endoplasmic reticulum stress, electron transport chain overload, adipose tissue hypoxia leading to increased secretion of adipokines and cytokines, as well as development of systemic insulin resistance [139,140].

We have demonstrated that in the round ligament of morbidly obese women (BMI of over 40), a specific hepatic adipose deposit highly developed during obesity, the level of ApoD is inversely correlated to the inflammatory profile of the patients. This is especially true with the expression of TNF-α and PAI-1, key markers of inflammation and thrombosis. The ApoD level in this specific adipose deposit is also positively associated with a better metabolic health, as noted by the evaluation of HOMA-IR and QUICKI indexes [141].

Interestingly, we have showed that HApoD-Tsg mice develop a non-inflammatory hepatic steatosis during middle age due to an upregulation of PPARγ, which increases CD36 expression, and lipid droplet expansion [74,142,143]. Usually, this degree of steatosis is associated with an increase in inflammation. However, this increase in inflammation was not observed in HApoD-Tsg mice [142]. In another study, we showed that this control of inflammation is age-dependent and mainly due to the altered prostaglandin metabolism [74]. Our data showed that ApoD overexpression redirects the ARA metabolism to increase the synthesis of 15d-PGJ_2_, an anti-inflammatory prostaglandin, by increasing the transcription of the lipocalin-type prostaglandin D synthase (Ptgds) gene. This increase in 15d-PGJ_2_ is concomitant with the onset of PPARγ upregulation and a decrease in NF-kB nuclear localization at 6 months old, preventing the establishment of a pro-inflammatory environment. The molecular mechanisms underlying the effect of ApoD remain, however, to be elucidated. Interestingly, in the hippocampus and cortex of mice lacking ApoD, Ptdgs is decreased [125], further pointing towards a role of ApoD in prostaglandin and ARA metabolisms.

The action of ApoD may be broader and affect other lipid mediator pathways. As HApoD-Tsg mice reached the age of 12 months, the levels of 15d-PGJ2 returned to normal. Despite the presence of an important hepatic steatosis, the inflammatory status of the mice was still unchanged. An interesting feature of their steatosis, however, was the high level of hepatic omega-3 fatty acids and consequently an improved ω-6/ω-3 ratio (lowered), which is known to decrease inflammation [74,144]. A role of ApoD may also include the upregulation in the production of anti-inflammatory omega-3-derived lipid mediators (*protectins*, *maresins* and *resolvins*). The normal inflammatory resolution process normally features a switch from the production of pro-inflammatory ARA-derived mediators (PGE2, LTB4) to its anti-inflammatory mediators (LXA4, LXB4, 15d-PGJ_2_) and finally to the production of the omega-3-derived lipid mediators (resolvins, maresins and protectins) [145]. A similar process may have occurred in the HApoD-Tsg mice, potentiated by ApoD’s action. These latter mediators were unfortunately not measured in that study. Considering ApoD’s effect on PLA2 and COX-2 activity, it would be interesting to ascertain whether ApoD inversely affect the lipoxygenases responsible for the synthesis of the omega-3-derived lipid mediators.

Confirming our observation, it was recently showed that mice overexpressing ApoD in their liver via adenoviral vector expression and fed a high-fat diet (to induce a chronic systemic inflammation) or injected with LPS (to induce acute inflammation) show lower circulating, hepatic, and adipose tissue inflammation markers TNF-α and IL-6 [40]. In this study, the authors also showed that ApoD overexpression increases plasmatic DHA and EPA (omega-3 fatty acids), and ARA levels and some of their metabolites. They also showed that hepatic overexpression of ApoD increased the anti-inflammatory lipid mediator sphingosine 1-phosphate, and on the contrary, decreased the proinflammatory adipokines osteopontin and autotaxin levels. Finally, ApoD was found to be positively correlated with ApoM, another lipocalin which is thought to be anti-inflammatory and beneficial for glucose tolerance and insulin resistance [40,146,147]. The authors also validate their findings in human, showing that the level of ApoD is inversely correlated with the expression of osteopontin and autotaxin while being positively correlated with ApoM levels [40].

## 4. Conclusions

In this review, we have shown the antioxidant and anti-inflammatory potential of ApoD (Table 1). ApoD is the gene most consistently upregulated in the brain with age [1]. This protein is also upregulated during pathologies, such as neurodegenerative diseases, OS, and inflammation. Using its Met93, mammalian ApoD can reduce some lipid hydroperoxides to non-reactive bioactive signaling molecules, and probably reduce other oxidants to decrease OS, as measured by the many studies mentioned using the OS markers 4HNE, MDA, F2-isoprostanes and 7-ketocholesterol. How and where ApoD works to reduce OS and inflammation is still largely unknown.

The internalization process in cells outside the nervous system is yet to be elucidated. We have shown that the level of ApoD’s accumulation in organs outside the CNS correlates with the level and degree of glycosylation of BSG [58]. Recently, it was demonstrated that ApoD can be internalized in BSG-KO U87 astrocytoma cells [62], pointing towards another mechanism of entry for this lipocalin. More studies are needed to show how ApoD can be internalized and if there exist differences in the internalization process in cells in the CNS and outside of it. In addition, the cellular localization of ApoD in different types of cells under specific conditions (OS and inflammation) has yet to reach a consensus (for example, in which type of cells and under what conditions ApoD is able to be in the nucleus).

Precise molecular demonstration of ApoD’s capacity to bind various lipids or other small lipophilic molecules to prevent the onset of LPO is needed. Moreover, more studies on the effect and cellular localization of ApoD on cells outside of the CNS are needed, since ApoD can exit the brain and accumulate in almost every organ [58]. Recently, it has been shown that adenovirus-mediated hepatic overexpression of ApoD reduces inflammation in adipose tissue, liver and, plasma in a model of a high-fat diet and LPS [40]. The exact mechanism through which ApoD decreases inflammation is not known, and would be of great interest to explore. We have been able to demonstrate that transgenic mice overexpressing the human ApoD will develop a non-inflammatory hepatic steatosis caused by an increase in fatty acid uptake due to increased hepatic CD36 expression, altered ARA metabolism, and amelioration of the ω-6/ω-3 ratio [74]. We hypothesized that ApoD could potentially bind various ω-3, bringing them to the liver where the accumulation of these PUFAs could create an anti-inflammatory environment. Although no ligand-binding assays have been carried out showing the interaction between omega-3s and ApoD, new evidence supports an interaction between the two, as the fraction of immunoprecipitated ApoD from the plasma of mice on LPS also contains DHA and EPA, both ω-3 [40].

Finally, ApoD’s capacity to sequester ARA in the cytosol and/or the cellular membrane should be clearly demonstrated, as should its capacity to alter ARA metabolism, either by modulating the expression implicated in ARA metabolism or by directly modulating their activity.

Since ApoD is a small protein implicated in many pathologies, and its mechanistic mode of action is largely unknown, its therapeutic potential has yet to be fully understood. For now, since it positively affects OS, downregulates inflammation, and also has relatively positive effects on the metabolic syndrome in some tissues, its potential as a therapeutic protein can be considered high, and its effects are worth exploring.

## Figures and Tables

**Figure 1 antioxidants-12-01027-f001:**
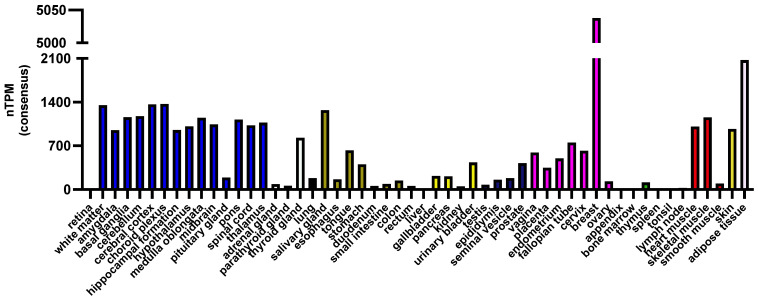
Normalized transcriptomic data consensus (nTPM) of Apolipoprotein D (ApoD) in humans by tissue. Data was taken from https://www.proteinatlas.org/ENSG00000189058-APOD/tissue (accessed on 26 April 2023).

**Figure 2 antioxidants-12-01027-f002:**
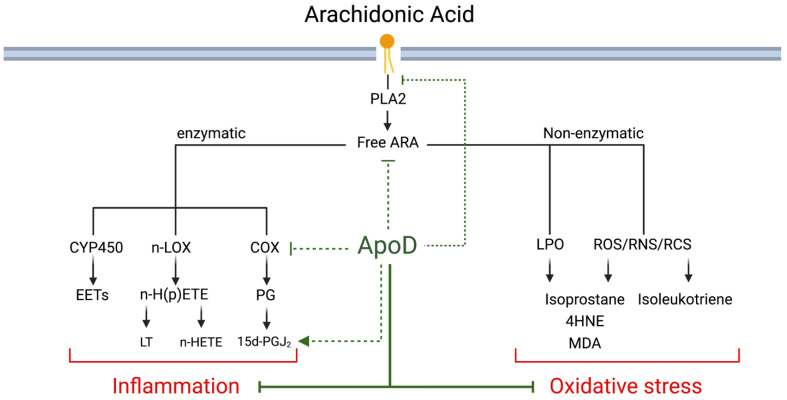
Effect of ApoD on arachidonic acid (ARA)-related inflammation and oxidative stress. ARA can be metabolized enzymatically (via CYP45, LOX or COX enzymes) to generate EETs, LTs (leukotrienes) and n-HETEs or prostaglandins (PG) or non-enzymatically through reactions with LPO or various reactive species to generate isoprostanes, isoleukotrienes, and 4HNE and MDA. ApoD alters PG metabolism by decreasing PLA2 activity, COX-2 expression, and by increasing the anti-inflammatory 15d-PGJ2. ApoD and can reduce H(p)ETEs to non-reactive HETEs via its Met93. ApoD can also decrease OS markers decreasing the deleterious lipid peroxidation of ARA (4HNE, MDA, F2-isoprostane), potentially by sequestering it directly. Dotted green lines show indirect or unknown mechanism of action that has a positive effect on either inflammation (reducing it) or OS (decreasing the markers identified).

**Figure 3 antioxidants-12-01027-f003:**
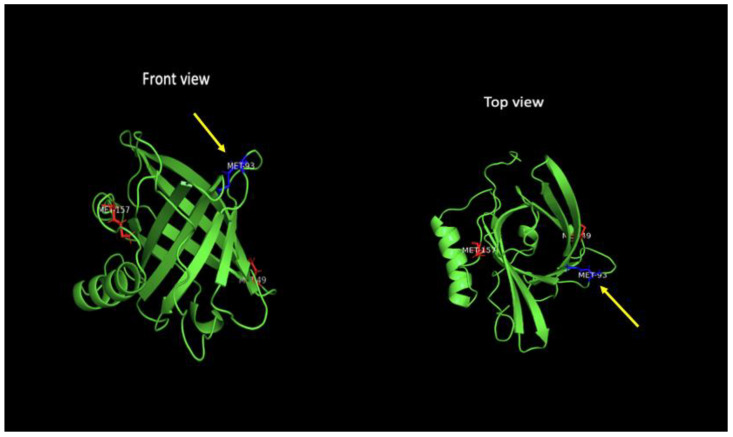
Three-dimensional representation of ApoD’s structure. The methionine93 residue (colored in blue) is responsible for the reduction of 5s-,12s- and 15s-H(p)ETEs, while the other two methionine (Met49 and Met157, in red) are not able to interact. Generated with PyMOL [126].

**Table 1 antioxidants-12-01027-t001:** Summary of studies carried out on Apolipoprotein D on various disorders (pathologies, oxidative stress, inflammation).

Model		Disorder	Effect	Source
** Human **		Parkinson	Increased ApoD expression in glial cells substantia nigra.	[3]
		Obesity	ApoD increases insulin sensitivity, decreases inflammation.	[141]
		Alzheimer	Increased ApoD expression in pyramidal neurons of the entorhinal cortex, ApoD dimers level correlates with levels of conjugated dienes in the hippocampus.	[4,10]
** Rodents **	HApoD-Tsg	inflammation	ApoD overexpression improved inflammation resorption, survival.	[22]
			ApoD overexpression increased latency to seizure, decreased inflammation and apoptosis.	[73]
			ApoD overexpression increases anti-inflammatory lipid mediators, ameliorated o-6/o-3 ratio, decreased inflammation (liver).	[74]
		Oxidative stress	ApoD overexpression decreased lipid peroxidation.	[8]
	ApoD-KO	Oxidative stress	Lack of ApoD increased lipid peroxidation in the brain, increased lipofuscin in the aging brain.	[8,125]
	HApoD-Tsg/ApoD-KO- APP/PS1	Alzheimer	Without ApoD: BACE1 level and plaque load increased. With ApoD overexpression: BACE1 level and plaque load decreased.	[100]
	ApoD overexpression in SR-BI/ApoE dKO	Atherosclerotic coronary artery disease	ApoD overexpression reduced myocardial infarction size.	[106]
	ApoD overexpression (Hepatic)	Obesity, LPS	lipid mediators are modulated, inflammation and osteopontin are decreased with ApoD overexpression.	[40]
** Drosophila **	Human ApoD, Nlaz, or Glaz overexpression	Oxidative stress	Overexpression of HApoD, Nlaz, or Glaz increased lifespan, survivability upon OS, decreased LPO observed in aging.	[54,108,110]
	Glial frataxin deficient coexpressing Glaz		Co-expression of Glaz in frataxin-deficient flies increases lifespan and survivability upon hyperoxia, restores aconitase activity.	[111]
	Nlaz or Glaz mutant		Lack of Nlaz or Glaz decreased lifespan, survival upon OS, increases LPO.	[54,107]
** Arabidopsis Thalia **	AtTIL overexpression/AtTIL-KO	Oxidative stress	AtTIL increases OS resistance.	[114]
	LNCP-null expressing HApoD		LCNP-KO with HApoD-expression decreases lipid peroxidation upon PQ treatment and have improved drought tolerance.	[115]
** Ex vivo/in vitro **	Human NPA fibroblast	Niemann–Pick disease Type A	Exogenous ApoD prevents lysosomal pH alkalinization, decreases lipid peroxidation and improves cell survival.	[64]
	HT-29 colorectal cancer cells	Colorectal cancer	Exogenous addition of ApoD to HT-29 cells promoted apoptosis upon paraquat-induced OS.	[9]
	ApoD-KO and WT astrocytes/neurons	Oxidative stress	ApoD from astrocytes decreases OS, increases survivability, protects lysosomes from alkalinization.	[55,60,121]
	Primary cultured rat myocardiocytes		ApoD protect primary cultured rat cardiomyocytes from hypoxia/reoxygenation injury.	[106]
	Rat hippocampal culture slices		ApoD decreased F2-isoprostane and 7-ketocholesterol.	[71]
	BmN cells		BmApoD1 decreased H_2_O_2_ OS and Actinomycin D-induced apoptosis.	[26]

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
