# Peer review of "Apolipoprotein D in Oxidative Stress and Inflammation"

_antioxidants, 2023, doi:10.3390/antiox12051027_

Round 1
Reviewer 1 Report
This comprehensive review is obviously written by experts in the field. The review is well structured and there is a logical flow of data. Minor comments: 1. The number of non-standard abbreviations should be drastically reduced. All these non-standard abbreviations undermine readability of the manuscript. 2. Please correct the layout of line 263-265. 3. Line 676. It is TNF-alpha (Greek letter). 4. I recognize that there is a therapeutic potential of apo D. However, gene therapy is never going to be safe enough for widespread systemic administration. Recombinant apo D could be envisaged in short term applications.Author Response
Dear reviewer,
Thank you for your suggestions.
1. The number of abbreviations has been lowered to make the text ''lighter'' and to remind readers of various abbreviations meaning. We have also completely abolished some abbreviation and replaced them by their full meaning in some parts of the text. For example, in section 3.1 Modulation of inflammation in the nervous system, we opted to remove PCI to write post crush injury and NCM for nerve crush model. We have also removed earlier on some abbreviations that come back only once to write their full meaning latter in the text.
2. and 3. Thank you for notifying us of our errors, which have been corrected.
4. This is a great point on which we agree. Random insertions into the genome can be very deleterious for an organism, and indeed recombinant ApoD treatment would be prefereable.
We have joined a copy of the revised manuscript for you to look at.
Sincerely, thank you for your comments and time spent revising our article.

Reviewer 2 Report
This is an interesting review article, addressing an area which clearly requires further investigation. Some thoughts for the authors are indicated below:
1. There are some ambiguous comments, not familiar to this reviewer: e.g. 'round tissue' relating to adipose tissue depots (Abstract)
2. There are some minor English language difficulties (see below)
3. The structure of the review seems to lose its way - I would prefer to see a strict numbering sequence adopted, so that the sub sections can be clearly identified
4. One obvious question is whether apoD localises in lipid rafts to bring about some of the cellular effects described?
5. It is not always clear whether the increases in apoD observed can be considered a causal part of the disease aetiology, or a response to pathology.
6. The authors should eliminate any speculation (or appearance of speculation), or clearly identify it as such. There are a number of statements that use 'could' or 'suggesting that'.
7. Is the preferential binding of ApoD to arachidonic acid solely dependent on reference [23]? This seems a key point, and should be evidenced more strongly
8. The ordering of the evidence on the function of apoD seems odd: to move from rodents to insects is counter-intuitive
9. The role of Met93 - it is not until the later sections that this is dissected, leaving the reader with a rather misleading focus on the function of this residue. Perhaps this could be dealt with earlier?
There are minor English language issues: e.g. line 64, lipocalin(s) requires plurality; line 173 'will' is the incorrect tense
Author Response
Dear reviewer,
Thank you for taking the time to read and comment our manuscript. We have revised your suggestions.
- There are some ambiguous comments, not familiar to this reviewer: e.g. 'round tissue' relating to adipose tissue depots (Abstract)
We have modified ''round tissue'' in the abstract to ''round ligament'', as this was a mistake on our part. Later in the manuscript, we explain that round ligament is an adipose tissue found on the liver in obese individual.
2. There are some minor English language difficulties (see below)
The mistakes were corrected.
3. The structure of the review seems to lose its way - I would prefer to see a strict numbering sequence adopted, so that the sub sections can be clearly identified
Thank you for this suggestion. The article was initially supposed to numbered, but during the process of internal modification of the structure of the paper, this was later abandonned for clarity's sake (mostly for us), and was never put back. We hope our new structure makes it more clear for you.
4. One obvious question is whether apoD localises in lipid rafts to bring about some of the cellular effects described?
No studies have demonstrated that. It is of great interest for the field to explore the more molecular mechanistic aspects of ApoD. It was demonstrated that ApoD can colocalize with some lipid raft markers such as Caveolin, but the link between these lipid rafts and the effect that ApoD brings has remain unexplored.
5. It is not always clear whether the increases in apoD observed can be considered a causal part of the disease aetiology, or a response to pathology.
This is our fault, as it was assumed but not explained that ApoD is beneficial in the context of pathologies. We have added a sentence very early on to make it clear that the upregulation of APOD is beneficial and does not contribute to the severity of pathologies (line 36-38).
6. The authors should eliminate any speculation (or appearance of speculation), or clearly identify it as such. There are a number of statements that use 'could' or 'suggesting that'.
Thank you for this very pertinent suggestion. It is easy while writing to know that our statements are speculations, but while reading it independently it might be confusing. We have added sentences throughout the texts, particularly where ''could'' and ''suggesting'' are to make it clear when we are speculating.
7. Is the preferential binding of ApoD to arachidonic acid solely dependent on reference [23]? This seems a key point, and should be evidenced more strongly
We have added another reference to make it more evident that ApoD binds strongly arachidonic acid. Lack of studies pertaining ApoD and ligand binding assay is a major shortfall in this particular field. We have a sentence in the conclusion where we explain more on this.
Althought there are little studies exploring ApoD ligand binding, there are a lot of studies making a direct link between ApoD and arachidonic acid levels or metabolism.
8. The ordering of the evidence on the function of apoD seems odd: to move from rodents to insects is counter-intuitive
This is true, and might be corrected when we rectified the structure of the paper to make the sections numbered. To make it more clear : The section ''2.2 animal studies'' contains ''2.2.1 rodents'' and ''2.2.2 insects''.
9. The role of Met93 - it is not until the later sections that this is dissected, leaving the reader with a rather misleading focus on the function of this residue. Perhaps this could be dealt with earlier?
This we previously discussed by our authors. We have now tried to make it more clear by directing the reader to the section ''2.5 in silico studies'' the first time Methionine 93 (met93) is mentioned. The idea was to make it more like funnel, starting from living studies (humans, rodents, insects, plants) to ex-vivo studies and to end with in silico studies with molecular dynamics simulations.
We thank you, sincerely for your comments, as they contribute to make this manuscript better. We hope you are satisfied with our revisions and get back to us soon.

Reviewer 3 Report
Report on the manuscript entitled „Apolipoprotein D in oxidative stress and inflammation” (antioxidants-2368411)
I read the text with a great pleasure and I will support its publication in Antioxidants.
Only one remark or suggestion. It must be strongly stressed in the text beginning from the abstract that the upregulation of APOD in several pathological conditions is not related to the situation that APOD may cause somehow those illnesses but this is a defense reaction of an organism. The sentence from the abstract “The APOD gene is upregulated in a number of pathologies, including Alzheimer’s disease, Parkinson’s disease, cancer, and hypothyroidism” when is read alone may being understood in a false way.
References: please unify the names of journals: in full or abbreviations. Now, it is a mix of them.
Author Response
Dear reviewer,
Thank you for your comments and suggestions.
- It must be strongly stressed in the text beginning from the abstract that the upregulation of APOD in several pathological conditions is not related to the situation that APOD may cause somehow those illnesses but this is a defense reaction of an organism
This is very true. It is our fault for making the assumption that the readers understand that ApoD's upregulation in pathologies is beneficial and not a cause of the pathologies. We have corrected this in the first paragraph of the introduction (line 36-38).
Thank you for your suggestions concerning the references. We have tried to correct them by using the MDPI style using Endnote, but the references do not change, as they are linked to the software and do not allow changes manually. Maybe the editors can manually change this before publishing?
Thank you for your comments, as they contribute to make this manuscript better. We hope our modifications are satisfactory.

Round 2
Reviewer 2 Report
The authors have answered queries to my satisfaction